# Information Extraction from Legal Wills: How Well Does GPT-4 Do?

**Alice Saebom Kwak[1], Cheonkam Jeong[1], Gaetano Vincent Forte[2],**
**Derek E. Bambauer[2], Clayton T. Morrison[3], Mihai Surdeanu[4]**

[1]Department of Linguistics, The University of Arizona
[2]James E. Rogers College of Law, The University of Arizona
[3]School of Information, The University of Arizona
[4]Department of Computer Science, The University of Arizona
{alicekwak, cheonkamjeong, fortegv, derekbambauer,
claytonm, msurdeanu}@arizona.edu

## Abstract

This work presents a manually annotated dataset for Information Extraction (IE) from legal wills, and relevant in-context learning experiments on the dataset. The dataset consists of entities, binary relations between the entities (e.g., relations between testator and beneficiary), and $n$-ary events (e.g., bequest) extracted from 45 legal wills from two US states. This dataset can serve as a foundation for downstream tasks in the legal domain. Another use case of this dataset is evaluating the performance of large language models (LLMs) on this IE task. We evaluated GPT-4 with our dataset to investigate its ability to extract information from legal wills. Our evaluation result demonstrates that the model is capable of handling the task reasonably well. When given instructions and examples as a prompt, GPT-4 shows decent performance for both entity extraction and relation extraction tasks. Nevertheless, the evaluation result also reveals that the model is not perfect. We observed inconsistent outputs (given a prompt) as well as prompt over-generalization.

## 1 Introduction

Wills are crucial legal documents that enable individuals to maintain control over their assets and ensure that their wishes are carried over after their death. Unlike most legal documents, wills are commonly written by/for lay people. Given the importance and ubiquity of wills, formalizing and developing legal procedures for understading, evaluating and executing wills is crucial. Extracting key information from legal wills is a prerequisite for understanding, evaluating and executing wills. Information extraction (IE) from legal wills serves as a foundation for many downstream tasks such as automatic will review and electronic will creation and execution.

Beyond social impact, will understanding is an important application of artificial intelligence (AI) for several reasons. First, will execution is an excellent use case for the flourishing smart contract platforms (National Science Foundation, 2022). Moreover, closer to natural language processing (NLP), information extraction from wills can serve as an important evaluation platform for large language models (LLMs). Lastly, given that most wills are not publicly available,[1] evaluating LLMs on IE from wills is less likely to suffer from contamination (Sainz et al., 2023) than other NLP tasks.

To enable such use cases, we introduce a dataset for IE from legal wills, and pertinent in-context learning experiments on this dataset. In particular, the main contributions of this work are as follows:

- We introduce a manually annotated dataset for IE from legal wills. Our dataset consists of 16,018 annotations of entities, relations, and events extracted from 45 legal wills from two US states: Tennessee and Idaho. The dataset can serve as a foundation for many downstream tasks concerning legal wills (e.g., automatic will review, electronic will creation, etc.)

- We evaluate GPT-4 with in-context learning on our dataset, in both in-domain (i.e., examples from the same state) and out-of-domain (OOD) (i.e., examples from another state) settings. The evaluation results demonstrate that GPT-4 is capable of handling the legal information extraction task in this in-context-learning setting. However, GPT-4 is not perfect: we observed inconsistent outputs (given a prompt) as well as prompt over-generalization.

---

[1]Court documents are in the public domain in the U.S., but wills are usually hidden behind pay-walled websites.

## 2 Related Work

### 2.1 IE from Legal Documents

There exist several IE datasets and models in legal domain. II et al. (2018) presents LexNLP, which is an open source Python package that can be used for natural language processing and information extraction from legal and regulatory texts. Chalkidis et al. (2017) introduces a manually annotated dataset for extracting key elements from contracts. Similarly, Tuggener et al. (2020) presents LEDGAR, a multi-label corpus for text classification of legal provisions in contracts. Chen et al. (2020) introduces a manually annotated dataset for named entity recognition and relation extraction in the Chinese legal domain, based on drug-related criminal judgment documents. Kirsch et al. (2020) proposes a probabilistic rule model for extracting information from court process documents. Martín-Chozas and Revenko (2021) describes a model extracting Hohfeld's deontic relations from legal texts. Hendrycks et al. (2021) introduces CUAD, an expert-annotated dataset for identifying key clauses from contracts. Cao et al. (2022) presents CAILIE 1.0, a Chinese legal information extraction dataset for larceny.

While IE from other legal documents tends to focus on entities and flat relations, IE from legal wills can involve information with more complexity, such as nested events.[2] Chalkidis et al. (2017) extracts various contract elements such as contract title, dates, relevant law articles, but it does not extract any relations among such elements nor extract events from the document. Chen et al. (2020), Kirsch et al. (2020), and Martín-Chozas and Revenko (2021) do extract relations, but the extracted relations are all flat ones. Chen et al. (2020) extracts four relations relevant to key criminal behavior, and these relations all have entities as their arguments. Likewise, the relations extracted from Kirsch et al. (2020) are all between entities (e.g., fine amount and fine units, etc.). The relations extracted from Martín-Chozas and Revenko (2021) all have entities (e.g., LegalEntity, LegalDocument, LegalConcept, and etc.) as their arguments.

In contrast, the target events in our work can have both entities and events as their arguments.

Take the *Direction* event in our work for instance. If what is being directed by a testator is another event (e.g., excuse a duty of the executor) in a will, the *Direction* event can have an *Excuse* event as its argument. Extracting nested events adds additional layer of complexity to the IE task. This higher level of complexity distinguishes our task (i.e., IE from legal wills) from IE in other legal domains.

There has been no prior endeavor to create an IE dataset from legal wills, despite their importance. There is a legal will dataset provided by Kwak et al. (2022), but its focus is on natural language inference rather than IE. Our work addresses this gap.

### 2.2 Evaluating LLMs on IE Tasks

Several studies have evaluated LLMs' performance on IE tasks in various domains. Agrawal et al. (2022) evaluated GPT-3 on diverse clinical extraction tasks in both a zero- and a few-shot learning settings and demonstrated that the model can handle the tasks well without training in the clinical domain. Stammbach et al. (2022) evaluated GPT-3 on the task of extracting character roles from narrative texts in a zero-shot question answering setting, and reported that the model is capable of handling the character role extraction task. Hanafi et al. (2022) evaluated GPT-3 on a text-pattern extraction task in a zero-shot setting and compared its performance with the rule-based model. He found that the rule-based model outperforms GPT-3 when GPT-3 is not given any prompts. Jimenez Gutierrez et al. (2022) evaluated GPT-3 on two biomedical IE tasks and compared its few-shot performance with the performance of smaller pre-trained language models (PLM) fine-tuned on the tasks. The result demonstrates that GPT-3 significantly underperforms a smaller PLM fined-tuned on the tasks. The previous studies suggest that LLMs are capable of handling IE tasks in a zero-shot or a few-shot setting with in-context learning. Nevertheless, there is no consensus on the performance of LLMs. Some reported the IE performance of LLMs in a positive way (e.g., Agrawal et al., 2022; Stammbach et al., 2022), while others reported it in a somewhat negative way (e.g., Hanafi et al., 2022; Jimenez Gutierrez et al., 2022). Further investigation is needed to verify LLMs' capability on IE tasks.

Recently, a few attempts have been made to evaluate LLMs on the legal IE task or closely related tasks. Barale et al. (2023) evaluates different ar-

---

[2]What we mean by "nested" versus "flat" is whether an event can have another event as its argument. If an event has another event as its argument, it is a nested event. On the other hand, if an event only has entities as its arguments, it is a flat event.

chitectures, including two BERT-based LLMs, on the legal IE task (i.e., extracting key information from refugee cases in Canada). Savelka (2023) evaluates GPT-4 on a semantic annotation of legal texts. However, the domain and the task of these previous works are distinct from those of our work. Barale et al. (2023) tests BERT-based LLMs on the IE from refugee cases, while our work tests GPT-4 on the IE from legal wills. Savelka (2023) evaluates GPT-4 as we do, but the task is different from ours (i.e., semantic annotation vs. IE). Hence, we evaluate GPT-4 on legal IE tasks with our dataset to investigate its capability of handling IE in this challenging domain.

## 3 Dataset

### 3.1 Data Collection

We collected 45 wills from Ancestry, which contains documents in the public domain[3]. The wills are from two US states (Tennessee and Idaho) and their execution dates range from 1978 to 2001. We performed an OCR on the collected wills to extract texts. More details on the OCR process can be found in Appendix A The will texts were anonymized with special tokens prior to the annotation process. Anonymization was done manually by our annotators.

### 3.2 Data Annotation

We annotated the data with the Label Studio Enterprise edition[4]. Annotation was performed by two annotators. They established a detailed annotation guideline to keep the consistency of their annotations (See Appendix C for the annotation guideline). Moreover, the two annotators had regular meetings to maintain the quality of the annotations. The inter-annotator agreement score was 74%, which was automatically calculated by the Label Studio with the basic matching function[5].

We annotated 26 types of entities, 18 types of binary relations, and 20 types of $n$-ary events, resulting in 16,108 annotations in total. With binary relation we extract relations between two entities (e.g., testator-beneficiary), while with $n$-ary events we extract events involving several arguments (e.g., bequest). Figure 1 shows an annotation example

---

[3]Court documents are in the public domain in the US.
[4]https://labelstud.io
[5]The Label Studio provides not only agreement score per token, but also the inter-annotator agreement. For further information, see https://docs.humansignal.com/guide/stats.html

---

**Entities:**
- testator: I, my, my
- trigger: hereby give, devise and bequeath, death
- asset: all of my property, real, personal or mixed
- beneficiary: [Person-2]
- condition: if living at the time of my death
- time: at the time of my death

**Relations:**
- testator-beneficiary:
  - testator: my
  - beneficiary: [Person-2]
- testator-asset:
  - testator: my
  - asset: all of my property, real, personal or mixed
- beneficiary-asset:
  - testator: [Person-2]
  - asset: all of my property, real, personal or mixed

**Events:**
- bequest:
  - testator: I
  - trigger: hereby give, devise and bequeath
  - asset: all of my property, real, personal or mixed
  - beneficiary: [Person-2]
- death:
  - trigger: death
  - testator: my

Figure 1: An example of an anonymized will statement, and corresponding annotations.

that includes entities, binary relations, and $n$-ary events. When deciding the annotation taxonomy, we consulted with a law professor and a law student. Appendix B contains the complete taxonomy of entities, relations, and events annotated.

## 4 Evaluation

Our dataset can be used to evaluate LLMs' performance on IE task. To this end, we used GPT-4[6] with in-context learning. When evaluating the model, we restricted the range of the task to extracting the four most-common entities (testator, beneficiary, asset, will) and the four most-common relations (testator-beneficiary, testator-asset, beneficiary-asset, testator-will) to reduce the complexity. Our dataset was randomly split into four sets: a training set (30 Tennessee wills), a development set (5 Tennessee wills), a test set (5 Tennessee wills), and an out-of-domain (OOD) set (5 Idaho wills). The training set served as an example pool for developing a prompt, and the development set was used to optimize the prompt. The evaluation was conducted on the test set and the OOD set.

We treated the wills from Idaho as the OOD set given the differences between the wills from Tennessee and the wills from Idaho. The most obvious

---

[6]https://openai.com/research/gpt-4

| Datasets | Precision | Recall | F1 |
|---|---|---|---|
| Test Entities | 0.65 (0.59 – 0.70) | 0.69 (0.64 – 0.73) | 0.67 (0.62 – 0.71) |
| Test Relations | 0.75 (0.75 – 0.93) | 0.76 (0.60 – 0.96) | 0.75 (0.69 – 0.90) |
| OOD Entities | 0.77 (0.70 – 0.83) | 0.74 (0.69 – 0.80) | 0.76 (0.71 – 0.80) |
| OOD Relations | 0.81 (0.70 – 0.89) | 0.85 (0.78 – 0.92) | 0.83 (0.76 – 0.88) |

Table 1: The result of evaluating GPT-4 on the legal IE task with in-context learning. We report precision, recall, and F1 scores with confidence intervals (calculated using bootstrap resampling with 10,000 samples) for entity and relation extraction. The "OOD" rows contain out-of-domain results, where we used examples from Tennessee on wills from Idaho.

difference between the two is that they are written to comply with different legal requirements, as Idaho and Tennessee have different probate codes. While Idaho is one of the states that adopted the entire Uniform Probate Code (UPC), which is a model law promulgated by the Uniform Law Commission, Tennessee only adopted a few provisions of the UPC. Tennessee's statutory framework is based upon state statutes, largely derived from the state's common law; it is, in other words, an example of the probate code heterogeneity that the UPC attempted to remediate. As they have different probate codes, the wills from Idaho and the wills from Tennessee must conform to different requirements. This can dictate the way the wills are formulated both substantively and stylistically.

We evaluated the model in an in-context learning setting where only a prompt with a few examples is given without fine-tuning the LLM. The prompt utilized in the evaluation was manually crafted using the training set and development set. The prompt consists of two parts: instruction and example. In the instruction part, a detailed instruction regarding the IE task is provided. In the example part, examples of human annotations are given to offer further guidance. After the instruction and the examples are provided, a will statement on which the model should perform the IE task is given with a brief instruction. For the relation extraction task, a set of entities extracted from the given will statement (by human annotators) are also provided. See Appendix D for the actual prompts used in our study.

The LLM outputs were compared with the human annotator's annotations. For the entity extraction task, we used exact text match as an evaluation metric, i.e., if the entire span of the extracted text does not exactly match the human annotation, it is considered incorrect. For the relation extraction task, we considered an extracted relation correct when any two mentions of the entity arguments match exactly and the correct type is produced. In

many cases, GPT-4 extracted multiple instances of a single relation, creating a large set of duplicates. These duplicates were ignored and removed before the evaluation.

## 5 Results

The evaluation results are summarized in Table 1. The table suggests that GPT-4 handles the IE task reasonably well. The F1 scores range from 0.67 to 0.83, which are decent given the strictness of the evaluation metric and the difficulty of the tasks. The scores from the entity extraction tasks are slightly lower than the ones from the relation extraction tasks (test: 0.67 vs. 0.75, OOD: 0.76 vs. 0.83). This difference is attributable to the rigorousness of the evaluation metric used for entity extraction task (i.e., exact text match for *all* entity mentions), rather than GPT-4's performance difference between the two tasks.

It is also noticeable that GPT-4 performed better on the OOD data than on the in-domain test data. This result suggests that the prompts created from our training partition can be transferred and used to extract information from OOD data (e.g., legal wills from the US states other than Tennessee).

## 6 Discussion

The evaluation result shows that GPT-4 is capable of handling the legal IE tasks without given training nor fine-tuning. However, the error analysis revealed two major issues of GPT-4's performance on IE tasks in a few-shot setting, which are 1) inconsistency and 2) overgeneralization of the prompt.

### 6.1 Inconsistency

GPT-4 is not capable of setting consistent criteria from the information given from the prompt. When not given training or fine-tuning, it failed to show consistent performance in various situations (e.g., whether to include possessive pronouns/quantifiers

| Error type | Test | OOD |
|---|---|---|
| Span difference | 143 (46%) | 66 (29%) |
| False positive | 91 (30%) | 75 (33%) |
| False negative | 71 (23%) | 77 (34%) |
| Human mistake | 2 (1%) | 8 (4%) |
| **Total** | 307 | 226 |

Table 2: Error analysis on the entity extraction task. For test data, span difference is the most common error type. For out-of-domain (OOD) data, all three types of error (except for human's mistake) are evenly present.

| Error type | Test | OOD |
|---|---|---|
| False positive | 16 (20%) | 22 (41%) |
| False negative | 53 (66%) | 34 (42%) |
| Human mistake | 11 (14%) | 9 (17%) |
| **Total** | 80 | 54 |

Table 3: Error analysis on the relation extraction task. For both test data and out-of-domain (OOD) data, the most common error type are false negatives. In OOD data, false positives are another major error type.

as part of the entity or not, whether to divide a list of entities into multiple small entities or to capture the list as a single entity, etc.). This inconsistency resulted in a high number of span difference errors in the entity extraction task (46% for test data and 29% for OOD data).

GPT-4 shows inconsistency in the relation extraction task as well. Unlike human annotators, who always extract a single instance of the relation between the same entities given a context, GPT-4 extracts a random number of instances of the relation between the same entities. It might extract all or none of them, or it can be somewhere in between. This arbitrary behavior of GPT-4 led to a considerable number of redundant extractions. These redundant extractions are not "wrong", so they were ignored when evaluating the model's performance. Nevertheless, this behavior can be problematic when someone plans to utilize the model for relation extraction tasks. As its way of extracting relations is arbitrary, one cannot anticipate its behavior based on its prior outputs nor process the output in a consistent way.

### 6.2 Overgeneralization of the Prompt

We witnessed many false positives (Test: 30% and OOD: 33% for the entity extraction task; Test: 20% and OOD: 41% for the relation extraction task) from the outputs generated by GPT-4. In the entity extraction task, we encountered cases where the model falsely extracted human entities (e.g., "Co-Executors") as beneficiaries, money-related entities (e.g., "tax", "expenses", "debts") as assets, and

document-related entities (e.g., "Tennessee Code Annotated, Section 35-50-110") as wills.

We witnessed the behavior of creating false positives in relation extraction as well. As explained in the previous section, we provided a gold set of entities as a part of the prompt for the relation extraction task, and thus did not expect to see many false positives. Nevertheless, GPT-4 did not limit itself to use only the provided set of entities but extracted additional entities from the given will text. These additionally extracted entities were mostly false positives (similar to the ones explained above), and thus the relations extracted based on these false positive entities were also false positives.

Given that this behavior does not occur when GPT-4 is asked to identify key elements from wills without given a prompt, it is very likely that this issue occurs due to GPT-4's overgeneralization of the prompt. Including examples for negative cases in the prompt might have prevented this issue from occurring.

## 7 Conclusion

We present an IE dataset consisting of 16,018 annotations from 45 legal wills. This dataset is valuable to both legal domain and AI domain, as it can serve as a foundation for several downstream tasks concerning legal wills. The dataset can also serve as an important evaluation platform for LLMs. We evaluated GPT-4 with in-context learning on our dataset, and demonstrated that the model can handle legal IE tasks reasonably well. However, the error analysis revealed two major issues of the GPT-4 in in-context learning setting, which are inconsistency and prompt overgeneralization. Further investigation is needed to mitigate these issues and enhance the performance of GPT-4 in in-context learning setting. Our open-access dataset is publicly available at: https://github.com/ml4ai/ie4wills/

### Limitations

The domain of our dataset is rather limited. It only contains English data, and the data came from two US states (i.e., Tennessee and Idaho) only. We plan to expand our dataset to other domains (e.g., adding more wills from other US states) in the near future. Also, the evaluation on LLM was only conducted on a subset of our dataset (i.e., four most common entities and four most common relations). It would be desirable to conduct an evaluation on the entire

data as a follow-up study to verify the findings from this study. Even though our dataset provides foundation for developing automatic information extraction model, this study does not present such a model. Further endeavor is needed to develop such a model, but it is beyond the scope of our current study.

## Ethics Statement

Our dataset contains legal wills collected from Ancestry. These legal wills are in public domain in the US, and we did not violate Ancestry's Terms and Conditions during the data collection process. All the collected wills were manually anonymized prior to annotation process.

## Acknowledgements

We thank the reviewers for their thoughtful comments and suggestions. We also appreciate Label Studio for providing access to Label Studio Enterprise Cloud Platform through their Academic Program. This work was partially supported by the National Science Foundation (NSF) under grant #2217215, and by University of Arizona's Provost Investment Fund. Mihai Surdeanu and Clayton Morrison declare a financial interest in lum.ai. This interest has been properly disclosed to the University of Arizona Institutional Review Committee and is managed in accordance with its conflict of interest policies.

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

## A  Legal Will Data Engineering Process with OCR

The legal will data we collected are saved in PDF format. In order to extract texts from the PDF files, we utilized two Python libraries: pdf2image and pytesseract.

We first converted the PDF files into PNG image files. After the conversion, texts were extracted from the images utilizing image_to_string function. Extracted texts were saved in a CSV file. The CSV file was manually reviewed by our annotators. Any errors were fixed during this review process. The code used for this process can be found from the following link: https://github.com/ml4ai/ie4wills/

## B  Taxonomy of entities, relations, and events

Our dataset contains 26 types of entities, 18 types of binary relations, and 20 types of *n*-ary events. See Tables below for the full taxonomy and its statistics.

### B.1  Entities

This section is formatted in the following way:
*Entity name*: a short description
ex) An example sentence showing all the relevant entities in **bold**.

1. *Testator*: a person who makes a will
   ex) **I**, **[Person-1]**, domiciled in Memphis, Tennessee, do make, publish and declare this to be **my** Last Will and Testament, hereby revoking all wills and codicils heretofore made by **me**.

2. *Trigger*: event trigger words. (extracted to be used in the *n*-ary event extractions)
   ex) A trigger for the *Nomination* event: I **name, nominate and appoint** my niece, [Person-2], Executor of this my will and estate, and direct that she be allowed to serve without bond.

3. *Condition*: a condition under which an event (e.g., will execution, bequest, etc.) occurs
   ex) I hereby give, devise and bequeath all of my property, real, personal or mixed, to [Person-2], **if living at the time of my death**.

4. *Beneficiary*: a person or entity (e.g., organization) that receives something from a will

ex) My personal representative shall be given credit on the final settlement for all disbursements made to **my beneficiaries** under the provisions of this power.

- Named beneficiary: a person or entity named in a will as a beneficiary
  ex) I will, devise and bequeath my house and lot to **[Person-16]** and **[Person-17]**.

5. *Will*: a legal document containing a person's wishes regarding the disposal of one's asset after death
   ex) I, [Person-1], domiciled in Memphis, Tennessee, do make, publish and declare **this** to be **my Last Will and Testament**, hereby revoking **all wills** and codicils heretofore made by me.

6. *Asset*: any money, personal property, or real estate owned by a testator
   ex) I hereby give, devise and bequeath **all of my property, real, personal or mixed**, to [Person-2], if living at the time of my death.

7. *Executor*: a person who executes a will (= personal representative)
   ex) I name, nominate and appoint **my daughter**, **[Person-3]**, **Executor** of this my will and estate, and direct that **she** be allowed to serve without bond.

8. *Witness*: a person witnessing a will
   ex) **We**, **[Person-5]** and **[Person-6]**, **the witnesses**, sign **our** names to this instrument, consisting of four pages, and being first duly sworn do hereby declare to the undersigned authority that the testatrix signs and executes this instrument as her last Will and that she signs it willingly, and that **each of us**, in the presence and hearing of the testatrix, hereby signs this Will as **witness** to the testatrix's signing, and that to the best of **our** knowledge the testatrix is eighteen (18) years of age or older, of sound mind, and under no constraint of undue influence.

9. *Time*: any expression denoting a particular point in time
   ex) I hereby give, devise and bequeath all of my property, real, personal or mixed, to [Person-2], if living **at the time of my death**.

10. *Duty*: any duty directed by a testator to fiduciaries (e.g., executors, trustees, guardians, or

conservators)

ex) I direct my Executor **to pay all my just debts and all funeral expenses**, which shall be probated, registered and allowed against my estate, as soon after my death as can conveniently be done.

11. *State*: any US state names

ex) I, [Person-1], domiciled in Memphis, **Tennessee**, do make, publish and declare this to be my Last Will and Testament, hereby revoking all wills and codicils heretofore made by me.

12. *Date*: any dates

ex) IN WITNESS WHEREOF, I have hereunto signed, published and declared this instrument as my Last Will and Testament, in Lauderdale County, Tennessee, **on this 12th day of June, 1989**

13. *County*: any US county names ex) IN WITNESS WHEREOF, I have hereunto signed, published and declared this instrument as my Last Will and Testament, in **Lauderdale County**, Tennessee, on this 12th day of June, 1989

14. *Right*: any rights authorized by a testator to fiduciaries (e.g., executors, trustees, guardians, or conservators)

ex) My personal representative shall have **the authority and discretion to buy or to sell or lease real property or any interest in real property which I may have** and **to use and apply the proceeds from a sale or lease to the payment of debts, taxes, and expenses of administration of my estate** and may **generally treat real property the same as personalty**.

15. *Expense*: any expenses

ex) I direct my Executor to pay all my just debts and **all funeral expenses**, which shall be probated, registered and allowed against my estate, as soon after my death as can conveniently be done.

16. *Debt*: any debts

ex) I direct my Executor to pay **all my just debts** and all funeral expenses, which shall be probated, registered and allowed against my estate, as soon after my death as can conveniently be done.

17. *Bond*: any bonds (usually probate bonds, which is a type of bond ordered and required by a court before they will appoint a person or entity as the personal representative of an estate)

ex) I name, nominate and appoint my daughter, [Person-3], Executor of this my will and estate, and direct that she be allowed to serve without **bond**.

18. *Codicil*: a testamentary or supplementary document that modifies or revokes a will or part of a will

ex) I, [Person-1], domiciled in Memphis, Tennessee, do make, publish and declare this to be my Last Will and Testament, hereby revoking all wills and **codicils** heretofore made by me.

19. *Trustee*: a person who manages a trust

ex) My entire estate, after payment of debts, taxes and expenses, shall be distributed to **the Trustee** of the Living Trust Agreement of [Person-1], entered into as of this same date of this my Last Will and Testament, to be administered and distributed under the terms of the trust created in the said Living Trust Agreement of [Person-1].

20. *Non-beneficiary*: a person who is excluded from being beneficiary

ex) In the event that **any other person or persons other than those herein named as my heirs** should seek to inherit from me and establish a right to so inherit by a final Decree of the Court of competent jurisdiction, then, in such event, I give and bequeath unto **such person or persons**, nothing.

21. *Tax*: any taxes

ex) I authorize my personal representative to pay from my general estate any interest which may accrue on debts or **taxes** due from my estate.

22. *Affidavit*: a legal statement sworn and signed by a testator and witnesses to confirm the validity of a will. It is usually attached to a will.

ex) IN WITNESS WHEREOF, I have executed this my Last Will at Ripley, Tennessee this 13th day of September, 2001 and request the attesting witnesses to make **the Affidavit** set out below.

23. *Notary public*: a person who is authorized by state government to witness the signing of important documents and administer oaths
    ex) Sworn to and subscribed before **me** on this, the 7th day of April, 1990.

24. *Trust*: a fiduciary arrangement that allows a trustee to hold assets on behalf of a beneficiary
    ex) I give, devise and bequeath the sum of five thousand dollars ($5,000.00) to my great-grandson [Person-11] to be held in **trust** for his future use and benefit until he reaches the age of twenty-five (25).

25. *Guardian*: a person who has a legal right and responsibility of taking care of someone who cannot takes care of themselves (usually a minor or an legally incompetent person)
    ex) If my personal representative determines that income or principal is payable to a minor or to a person under mental or physical disability, whether or not adjudicated, then my fiduciary shall have the discretion to either make payments directly to the beneficiary, to **the legally appointed guardian** or conservator, or to distribute and pay such amounts directly for the benefit of such beneficiary.

26. *Conservator*: a person who handles the financial and personal affairs who cannot handles such affairs by themselves (usually a minor or an legally incompetent person)
    ex) If my personal representative determines that income or principal is payable to a minor or to a person under mental or physical disability, whether or not adjudicated, then my fiduciary shall have the discretion to either make payments directly to the beneficiary, to the legally appointed guardian or **conservator**, or to distribute and pay such amounts directly for the benefit of such beneficiary.

## B.2  Relations

This section is formatted in the following way:
*Relation name*: a short description
ex) An example of the relation being explained followed by an example sentence.
(e.g., The relation between *entity 1* and *entity 2* in "an example sentence showing the entities participating in the relation being exemplified (i.e., *entity 1* and *entity 2*) in **bold**.")

| Entity types | Number |
|---|---|
| Testator | 1904 |
| Trigger | 1293 |
| Condition | 727 |
| Beneficiary | 56 |
| Named Beneficiary | 588 |
| Will | 545 |
| Asset | 525 |
| Executor | 465 |
| Witness | 315 |
| Time | 198 |
| Duty | 139 |
| State | 114 |
| Date | 108 |
| County | 85 |
| Right | 79 |
| Expense | 70 |
| Debt | 64 |
| Bond | 63 |
| Codicil | 34 |
| Trustee | 27 |
| Non-beneficiary | 25 |
| Tax | 20 |
| Affidavit | 18 |
| Notary public | 17 |
| Trust | 10 |
| Guardian | 5 |
| Conservator | 3 |
| **Total** | **7497** |

Table 4: The entitie types contained in the dataset and their statistics.

1. *Coreference resolution*: the relation between entities referring to the same real-world entity
   ex) The relation between *I* and *my* in "**I** name, nominate and appoint **my** niece, [Person-2], Executor of this **my** will and estate, and direct that she be allowed to serve without bond."

2. *Beneficiary-Asset*: the relation between a beneficiary and asset
   ex) The relation between *[Person-2]* (Beneficiary) and *all of my property, real, personal, or mixed* (Asset) in "I hereby give, devise and bequeath **all of my property, real, personal or mixed**, to **[Person-2]**, if living at the time of my death."

3. *Testator-Asset*: the relation between a testator and asset
   ex) The relation between *I* (Testator) and *all of my property, real, personal, or mixed* (Asset) in "**I** hereby give, devise and bequeath **all of my property, real, personal or mixed**, to [Person-2], if living at the time of my death."

4. *Testator-Beneficiary*: the relation between a testator and a beneficiary
   ex) The relation between *I* (Testator) and *[Person-2]* (Asset) in "**I** hereby give, devise

and bequeath all of my property, real, personal or mixed, to **[Person-2]**, if living at the time of my death."

5. *Testator-Will*: the relation between a testator and a will
ex) The relation between *I* (Testator) and *all former wills* in "**I** hereby expressly revoke **all former wills** and codicils by me at any time heretofore made."

6. *Testator-Executor*: the relation between a testator and an executor
ex) The relation between *I* and *my niece* in "**I** name, nominate and appoint **my niece**, [Person-2], Executor of this my will and estate, and direct that she be allowed to serve without bond."

7. *Inclusion*: the relation between an entity and another entity in which the former one includes the latter one
ex) The relation between *All the rest of my property* and *any furnishings inside the house, any money I have, and any other property of any kind* in "**All the rest of my property**, including **any furnishings inside the house, any money I have, and any other property of any kind**, I will and bequeath to [Person-6] of Maury City, Tennessee."

8. *Competence*: the relation between a testator and a condition that shows the competence of the testator
ex) The relation between *I* and *being of sound and disposing mind and memory* in "**I**, [Person-1], of Gates, Lauderdale County, Tennessee, **being of sound and disposing mind and memory**, do hereby make, publish, and declare this instrument as my LAST WILL AND TESTAMENT, hereby revoking all wills and codicils to wills heretofore made by me."

9. *Parent-Child*: the relation between a parent and a child
ex) The relation between *I* and *my son* in "**I** direct that my household furnishings and contents be divided in kind equally between **my son**, [Person-2], and my daughter-in-law, [Person-3]."

10. *Witness-Testator*: the relation between a witness and a testator
ex) The relation between *We* and *[Person-1]* in "**We**, the undersigned and subscribing witnesses, do hereby certify that we witnessed the foregoing Last Will and Testament of **[Person-1]**, at her request, in her presence, and in the presence of each other, and that she signed the same in our presence, and in the presence of each of us, declaring the same to be her Last Will and Testament."

11. *Witness-Will*: the relation between a witness and a will
ex) The relation between *We* and *the foregoing Last Will and Testament* in "**We**, the undersigned and subscribing witnesses, do hereby certify that we witnessed **the foregoing Last Will and Testament** of [Person-1], at her request, in her presence, and in the presence of each other, and that she signed the same in our presence, and in the presence of each of us, declaring the same to be her Last Will and Testament."

12. *Spouse-Spouse*: spousal relations
ex) The relation between *I* and *my husband* in "**I** appoint **my husband**, [Person-3], to be Executor of this my Last Will and Testament and direct that no security be required of him as such."

13. *Beneficiary-Executor*: the relation between a beneficiary and an executor
ex) The relation between *my personal representative* and *my beneficiaries* in "Beginning immediately after my death, **my personal representative** may begin making payments to **my beneficiaries** during the administration of my estate and need not wait for the final settlement of the estate."

14. *Beneficiary-Will*: the relation between a beneficiary and a will
ex) The relation between *this Will* and *any recipient* in "All inheritance, estate, and similar taxes assessed by reason of my death shall be paid in respect of all property and interest required to be included in my estate for tax purposes whether the same passes under **this Will** or otherwise and without contribution from **any recipient** of any property required to be included for tax purposes."

15. *Testator-Codicil*: the relation between a testator and a codicil

ex) The relation between *I* and *codicils* in "**I**, [Person-1], domiciled in Memphis, Tennessee, do make, publish and declare this to be my Last Will and Testament, hereby revoking all wills and **codicils** heretofore made by me."

16. *Testator-Trustee*: the relation between a testator and a trustee
    ex) The relation between *My* and *the Trustee* in "**My** entire estate, after payment of debts, taxes and expenses, shall be distributed to **the Trustee** of the Living Trust Agreement of [Person-1], entered into as of this same date of this my Last Will and Testament, to be administered and distributed under the terms of the trust created in the said Living Trust Agreement of [Person-1]."

17. *Testator-Guardian*: the relation between a testator and a guardian
    ex) The relation between *my* and *the legally appointed guardian* in "If **my** personal representative determines that income or principal is payable to a minor or to a person under mental or physical disability, whether or not adjudicated, then my fiduciary shall have the discretion to either make payments directly to the beneficiary, to **the legally appointed guardian** or conservator, or to distribute and pay such amounts directly for the benefit of such beneficiary."

18. *Testator-Conservator*: the relation between a testator and a conservator
    ex) The relation between *my* and *conservator* in "If **my** personal representative determines that income or principal is payable to a minor or to a person under mental or physical disability, whether or not adjudicated, then my fiduciary shall have the discretion to either make payments directly to the beneficiary, to the legally appointed guardian or **conservator**, or to distribute and pay such amounts directly for the benefit of such beneficiary."

### B.3 Events

This section is formatted in the following way:
*Event name*: a short description
ex) An example sentence

- Argument name: "argument example excerpted from the example sentence above"

| Relation types | Number |
|---|---|
| Coreference resolution | 2560 |
| Beneficiary-Asset | 376 |
| Testator-Asset | 361 |
| Testator-Beneficiary | 275 |
| Testator-Will | 245 |
| Testator-Executor | 189 |
| Inclusion | 165 |
| Competence | 156 |
| Parent-Child | 87 |
| Witness-Testator | 57 |
| Witness-Will | 49 |
| Spouse-Spouse | 37 |
| Beneficiary-Executor | 27 |
| Beneficiary-Will | 18 |
| Testator-Codicil | 10 |
| Testator-Trustee | 7 |
| Testator-Guardian | 3 |
| Testator-Conservator | 1 |
| **Total** | 4623 |

Table 5: The relation types contained in the dataset and their statistics.

1. *Bequest*: an event in which a testator bequeath asset to a beneficiary
   ex) I hereby give, devise and bequeath all of my property, real, personal or mixed, to [Person-2], if living at the time of my death.

   - Trigger: "hereby give, devise and bequeath"
   - Testator: "I"
   - Asset: "all of my property, real, personal or mixed"
   - Beneficiary: "[Person-2]"
   - Condition: "if living at the time of my death"

2. *Sign will*: an event in which a testator or a witness signs a will
   ex) IN WITNESS WHEREOF, I have hereunto signed, published and declared this instrument as my Last Will and Testament, in Lauderdale County, Tennessee, on this 11 day of April, 1994.

   - Trigger: "have hereunto signed"
   - Testator: "I"
   - Will: "this instrument"
   - Date: "on this 11 day of April, 1994"
   - Condition: "IN WITNESS WHEREOF"

3. *Direction*: an event in which a testator gives direction to a fiduciary
   ex) I direct that my Executrix not be required to make an accounting to the Court.

   - Trigger: "direct"

- Testator: "I"
- Directed event:
  - Excuse: "my Executrix not be required to make an accounting to the Court"

4. *Will creation*: an event in which a testator creates a will
   ex) I, [Person-1], an adult resident citizen of [Address-1], Lauderdale County, Tennessee, being of sound and disposing mind, memory and understanding, do hereby make, declare and publish this instrument as my Last Will and Testament, expressly revoking any and all testamentary dispositions heretofore made by me.
   - Trigger: "do hereby make, declare and publish"
   - Testator: "I"
   - Will: "this instrument"

5. *Attestation*: an event in which a witness attests the validity of a will
   ex) We, the undersigned subscribing witnesses, do hereby certify that we witnessed the foregoing Last Will and Testament of [Person-1], at her request, in her presence and in the presence of each other, and that she signed the same in our presence, and in the presence of each of us, declaring the same to be her Last Will and Testament. This 11 day of April, 1994.
   - Trigger: "do hereby certify"
   - Witness: "We"
   - Attested events:
     - Attestation: "we witnessed the foregoing Last Will and Testament of [Person-1], at her request, in her presence and in the presence of each other"
     - Sign will: "she signed the same in our presence, and in the presence of each of us"
   - Date: "This 11 day of April, 1994"

6. *Excuse*: an event in which a testator excuses a fiduciary from a duty
   ex) I name, nominate and appoint my daughter, [Person-3], Executor of this my will and estate, and direct that she be allowed to serve without bond.
   - Trigger: "without"
   - Testator: "I"
   - Executor: "she"
   - Bond: "bond"

7. *Authorization*: an event in which a testator authorizes a fiduciary to a right
   ex) I further give to my Personal Representative all of the powers of a Personal Representative under the laws of the state of Idaho as now in effect and as may hereafter be amended.
   - Trigger: "further give"
   - Testator: "I"
   - Executor: "my Personal Representative"
   - Right: "all of the powers of a Personal Representative"
   - Condition: "under the laws of the state of Idaho as now in effect and as may hereafter be amended"

8. *Nomination*: an event in which a testator nominates a fiduciary
   ex) I name, nominate and appoint my daughter, [Person-3], Executor of this my will and estate, and direct that she be allowed to serve without bond.
   - Trigger: "name, nominate and appoint"
   - Testator: "I"
   - Executor: "my daughter"

9. *Death*: an event in which any entity (e.g., testator, beneficiary, executor, etc.) dies
   ex) In the event that my said granddaughter does not survive me, then I hereby give, devise and bequeath said property to my great grandson, [Person-3]
   - Trigger: "does not survive"
   - Beneficiary: "my said granddaughter"

10. *Revocation*: an event in which a testator revokes a will or a codicil
    ex) I, [Person-1], of Gates, Lauderdale County, Tennessee, being of sound and disposing mind and memory, do hereby make, publish, and declare this instrument as my LAST WILL AND TESTAMENT, hereby revoking all wills and codicils to wills heretofore made by me.
    - Trigger: "hereby revoking"
    - Testator: "my"
    - Will: "all wills"

- Codicil: "codicils to wills"

11. *Probate*: an event in which a will or any part of the will is probated
   ex) I hereby direct my Executor to pay all of my just debts, funeral expenses, taxes and other expenses, which shall be probated, registered and allowed against my estate as soon after my death as can be conveniently done.

   - Trigger: "shall be probated"
   - Debt: "all of my just debts"
   - Expense: "funeral expenses"
   - Tax: "taxes"
   - Expense: "other expenses"
   - Condition: "against my estate"
   - Time: "as soon after my death as can be conveniently done"

12. *Codicil*: an event in which a codicil is made
   ex) I, [Person-1], of Gates, Lauderdale County, Tennessee, being of sound and disposing mind and memory, do hereby make, publish, and declare this instrument as my LAST WILL AND TESTAMENT, hereby revoking all wills and codicils to wills heretofore made by me.

   - Trigger: "made"
   - Testator: "me"
   - Codicil: "codicils"
   - Time: "heretofore"

13. *Disqualification*: an event in which a beneficiary or a fiduciary is disqualified
   ex) If, for any reason, [Person-3] is unwilling or unable to serve in this capacity, then I nominate and appoint his daughter, [Person-5], to serve in his place as Co-Executor without bond.

   - Trigger: "is unwilling or unable to serve"
   - Executor: "[Person-3]"

14. *Removal*: an event in which a beneficiary is removed from the will
   ex) I further direct that if any beneficiary under this Will should contest the terms of this Will they shall receive nothing by the terms of this Will and that share which they would have received shall be divided among the remaining beneficiaries of my estate in the same manner as though they had predeceased me without issue.

- Trigger: "shall receive nothing"
- Beneficiary: "they"
- Condition: "if any beneficiary under this Will should contest the terms of this Will"

15. *Give*: an event in which a testator gives a compensation to a fiduciary
   ex) I direct that my Executor shall receive a fee for his services of five (5) percent of my net estate after the above specific bequests are made.

   - Trigger: "shall receive"
   - Executor: "my Executor"
   - Asset: "five (5) percent of my net estate"
   - Time: "after the above specific bequests are made"

16. *Non probate instrument creation*: an event in which a non probate instrument (e.g., trust) is created
   ex) I give, devise and bequeath the sum of five thousand dollars ($5,000.00) to my great-grandson [Person-11] to be held in trust for his future use and benefit until he reaches the age of twenty-five (25).

   - Trigger: "be held"
   - Asset: "the sum of five thousand dollars ($5,000.00)
   - Trust: "trust"

17. *Renunciation*: an event in which a fiduciary renounces
   ex) If, for any reason, [Person-3] is unwilling or unable to serve in this capacity, then I nominate and appoint his daughter, [Person-5], to serve in his place as Co-Executor without bond.

   - Trigger: "is unwilling"
   - Executor: "[Person-3]"

18. *Notarization*: an event in which an affidavit is notarized by a notary public
   ex) SWORN TO before me this September 14, 2001. NOTARY PUBLIC My Commission expires: 6/19/06

   - Trigger: "SWORN"
   - Notary public: "me"
   - Date: "this September 14, 2001"

19. *Birth*: an event in which a beneficiary is born
    ex) I have two children: [Person-2], born [Date-1], and [Person-3], born [Date-2].

    - Trigger: "born"
    - Beneficiary: "[Person-2]"
    - Date: "[Date-1]"

    - Trigger: "born"
    - Beneficiary: "[Person-3]"
    - Date: "[Date-2]"

20. *Residual*: an event in which asset becomes residuary estate
    ex) If the gift of any item of property under this Article 1 fails or lapses, such property shall become a part of my residuary estate and shall be distributed as provided in Article 2.

    - Trigger: "shall become a part of my residuary estate"
    - Asset: "such property"
    - Condition: "If the gift of any item of property under this Article 1 fails or lapses"

| Event types | Number |
|---|---|
| Bequest | 756 |
| Sign will | 542 |
| Direction | 530 |
| Will creation | 472 |
| Attestation | 400 |
| Authorization | 249 |
| Excuse | 247 |
| Nomination | 208 |
| Death | 156 |
| Revocation | 105 |
| Probate | 91 |
| Codicil | 44 |
| Disqualification | 31 |
| Removal | 20 |
| Give | 11 |
| Non probate instrument creation | 11 |
| Renunciation | 11 |
| Notarization | 6 |
| Birth | 4 |
| Residual | 4 |
| **Total** | 3898 |

Table 6: The event types contained in the dataset and their statistics.

## C  Annotation Guideline

If you have any questions that are not already addressed in the guideline below, please keep them as separate notes so that we can discuss together and add them to the guideline later.

### C.1  Entity

- Check if all the entities are annotated with correct labels (see the taxonomy to check the full entity list)

- Use the most specific label when there's multiple possible labels
  ex) use *NAMED_BENEFICIARY* instead of *BENEFICIARY* when applicable.[7]

- When there's a priority among multiple beneficiaries/executors, capture the priority by using numbered beneficiary labels[8]
  ex) I designate [Person-1] as an executor. If [Person-1] cannot be an executor, then [Person-2] should be an executor.

  - In the sentence above, [Person-1] has a priority over [Person-2]. So we'll want to use *Executor1* for annotating [person-1] and *Executor2* for annotating [person-2].
  - In other cases, we'll just use the *Executor* label.
  - This applies to annotating *Beneficiary* as well.

- When annotating entities, include quantifiers (all, some, any, both, etc.) and articles (a, the, this, that)

- Label only what's present in the sentence. Don't worry about the entities omitted from the sentence

- Any special tokens (e.g., [Person-1]) should be captured as a single entity, including brackets

- Event trigger words (verbs/nouns triggering events) should be annotated with the "TRIGGER" tag

---

[7]When annotating the dataset, "beneficiary" and "named beneficiary" was distinguished. However, during the evaluation process, this distinction was ignored and both were treated as "beneficiary".

[8]When annotating the dataset, the priority among the same entities (e.g., executor or beneficiary) was marked with numbers. However, during the evaluation process, priority numbering was ignored.

- When multiple trigger words are listed for a single event, annotate the whole trigger words as a single trigger entity
  ex) "make, declare, and publish" as a single event trigger

- Any date should be annotated as a single date entity, including any preposition (e.g., "on") and a year. ex) "on this 12th day of June, 1989" as a single date entity

- For annotating "excuse" event, use negative word (e.g., "no", "without", etc.) as a trigger

- Include "to" when annotating verb phrases
  ex) to pay all my just debts

- During the review phase, make sure to check if the entities that were introduced later in the annotation phase (e.g., *Condition* and *Non-beneficiary*) were properly annotated

- All of the following should be annotated as conditions:

  - Competence: age or mental capacity of a testator
  - Conditionals: if-clauses, clauses starting with "In the event..."
  - Witnessing requirements: "at his/her request", "in his/her presence", "in the presence of each other", etc.
  - Any other phrases that might count as conditions:
    * "IN WITNESS WHEREOF"
    * "In fee simple"
    * "Equally or in equal amount"
    * "For X period of time"
    * "Without bond"

- Use the label *Time* only when the entity is temporal expression. Expressions such as "at my/her/his request" should be annotated as conditions.

- When annotating assets, annotate the whole phrase as one chunk. In the example below, the whole phrase ("my peacock collection, my humming birds, my sewing machine my jewelry armoire and all personal items (clothing, shoes coats etc.) which she may like") should be annotated as one entity.
  ex) I will, give, devise and bequeath unto my

very dear friend, [Person-7] **my peacock collection, my humming birds, my sewing machine, my jewelry armoire and all personal items (clothing, shoes coats etc.) which she may like**.

## C.2 Relations

- Check if all the relations have been annotated and correctly labeled (see the taxonomy to check the full relation list)

- Be mindful with the order of the entities being connected. The entities that are listed first in the relation label should be the starting point and those that are listed second in the relation label should be the ending point of the arrow connecting two entities
ex) If we're annotating "Testator-Beneficiary", the entity labeled "Testator" should be the starting point and the entity labeled "Beneficiary" should be the ending point.

- Coreference resolution should pair the most closely located entities referring to the same real-world entity within a sentence

- When there are multiple instances of entities participating in a relation, connect the most closely located instances and disregard all the other ones. We only need to mark the relation between the same real-world entities once, not multiple times. Coreference resolution will take care of this.
ex) I name, nominate and appoint my niece, [Person-2], Executor of this my will and estate, and direct that she be allowed to serve without bond.

  - If we are annotating the "Testator-Executor" relation in this example, we'll connect "my" (Testator) and "my niece" (Executor) as they are the most closely located ones to each other.
  - We don't need to connect other instances, as we already annotated their relation once. (e.g., We don't need to connect "I" and "[Person-2]")

## C.3 Events

- Check if all the events have been annotated and correctly labeled (see the taxonomy to check the full event list)

- Be mindful with the direction of the arrow connecting a trigger and arguments. It should ALWAYS start from a trigger and then connected to other arguments.

- When there's a need for using a specific event as an argument of another event (e.g., excuse, direction, authorization, and any other nested events), use the event's trigger as the argument.
ex) [...] **direct** that she be allowed to serve **without** bond.

  - Here, *Direction* event will have *Excuse* event as one of its argument. In this case, the trigger of the *Excuse* event (i.e., "without") will be connected to the trigger of the *Direction* event to show that the *Excuse* event is one of the arguments of the *Direction* event.

- Don't forget to connect all the *Time* and *Condition* entities to the relevant events.

- We will keep *Death* as an event. Any nouns/verbs relevant to death (e.g., "death", "die", "does not survive", etc.) should be annotated as triggers and the person involved in that event (i.e., a person who dies or is dead) will be connected to the trigger

# D   Evaluation Prompts

Figure 2 and 3 below show the actual prompts used to evaluate GPT-4 on legal information extraction tasks. The prompts consist of two parts: instructions and examples. After the prompt, a will text to perform information extraction on is given.

## D.1   Entity Extraction

---

%%Instructions: Your task is to identify all the instances (including pronouns) of TESTATOR, BENEFICIARY, WILL, and ASSET from the will statement provided. Refer to the example and the extracted information below as a guide to accurately complete the task. Fill in the relevant information for each category mentioned.

---

%%Example:
Will Text: ARTICLE 3. All the rest. residue and remainder of the property which I may own at my death, real, personal or mixed, tangible or intangible, of whatsoever nature and wheresoever situated including all property which I may acquire or become entitled to after the execution of this Will I bequeath and devise to my husband, [Person-3], if he be living at my death, and if he is not living at my death, I bequeath and devise this property to my son [Person-2].
---
Annotation Text: I
Start Index: 69
End Index: 70
Annotation Labels: ['TESTATOR']
---
… (18 more entities are given as examples)

---

%%Extract the relevant information from the following will statement and complete the form provided below:
Will Text: I, [Person-1] of the [Address-1], County of Cassia and State of Idaho, of the age of 72 years, being of sound mind and memory, and not acting under duress, menace, fraud or undue influence of any person whomsoever, do make, publish, and declare this my Last Will and Testament in manner and form following, to-wit:
---

---

Figure 2: A prompt used to evaluate the GPT-4 on legal entity extraction task.

## D.2   Relation Extraction

---

%%Instructions: Your task is to identify relations between TESTATOR and BENEFICIARY, TESTATOR and ASSET, BENEFICIARY and ASSET, and TESTATOR and WILL from the will statement provided. Refer to the example, entities and the extracted relations below as a guide to accurately complete the task. Fill in the relevant information for each relation category mentioned.

---

%%Example:
Will Text: ARTICLE 3. All the rest. residue and remainder of the property which I may own at my death, real, personal or mixed, tangible or intangible, of whatsoever nature and wheresoever situated including all property which I may acquire or become entitled to after the execution of this Will I bequeath and devise to my husband, [Person-3], if he be living at my death, and if he is not living at my death, I bequeath and devise this property to my son [Person-2].
---
Entity ID: jiZ94GhpJ2
Annotation Text: I
Start Index: 69
End Index: 70
Annotation Labels: ['TESTATOR']
---
… (9 more entities are given)
---
Relation Labels: ['TESTATOR-BENEFICIARY']
To_id: p-ge7sWfXg
From_id: r1PFNvb4TP
---
… (7 more relations are given)

---

%%Extract the relevant information from the following will statement and complete the form provided below:
Will Text: I, [Person-1] of the [Address-1], County of Cassia and State of Idaho, of the age of 72 years, being of sound mind and memory, and not acting under duress, menace, fraud or undue influence of any person whomsoever, do make, publish, and declare this my Last Will and Testament in manner and form following, to-wit:
---
Entity ID: i7r52xPaxP
Annotation Text: I
Start Index: 0
End Index: 1
Annotation Labels: ['TESTATOR']
---
… (4 more entities are given)
---

---

Figure 3: A prompt used to evaluate the GPT-4 on legal relation extraction task.