# OpenReview forum: "Information Extraction from Legal Wills: How Well Does GPT-4 Do?"
_EMNLP/2023/Conference — EMNLP 2023 Findings_

### Official Review · Reviewer_ZHWR · 2023-07-29

**Soundness:** 3

**Excitement:**

3: Ambivalent: It has merits (e.g., it reports state-of-the-art results, the idea is nice), but there are key weaknesses (e.g., it describes incremental work), and it can significantly benefit from another round of revision. However, I won't object to accepting it if my co-reviewers champion it.

**Missing References:**

Overall no important reference is missing, but here could be a few related papers that are not cited as related work:
- Tax Law Entailment, at NLLP workshop 2020: https://arxiv.org/abs/2005.05257
- LexNLP: https://arxiv.org/abs/1806.03688
- Legal NLP review: https://aclanthology.org/2020.acl-main.466/

**Paper Topic And Main Contributions:**

The short-paper present a new dataset for legal NLP (legal wills) and manually annotated data for the task of entity extraction and relation extraction.

**Questions For The Authors:**

- How do you explain that results are significantly higher for OOD data in table 1?
- Same question for table 2 span difference 46% vs. 29% for OOD data: how can this difference be explained?

**Reasons To Accept:**

- There is generally a lack of resources in legal NLP, and introducing new datasets that are ready to be used by the community is essential to develop the field.
- To the best of my knowledge, this is the first dataset containing wills.
- The authors gather about 16, 000 gold standard annotations, over 45 documents, which is a reasonable size for a legal dataset.
- The authors focus on the task of information extraction and linking, which are tasks that are commonly performed in the legal workload and that are often cited by legal professionals as ones that could be sped up using AI capabilities.
- Paragraph 2.2 compare the legal domain with other domains requiring specific expertise such a healthcare IE and few-shot learning. This parallel between domain is very interesting since there are a number of shared challenges such as specific vocabulary and contextual meaning. An interesting addition to this paragraph would be to point out those similarities.
- By performing experiment on out-of-domain data (documents from another state), the authors show that it is transferable at least to some extent within the USA.
- Paragraph 6.1 and 6.2 include a precise analysis of the models errors in the context of legal IE by exposing both inconsistencies and overgeneralization. This is highly valuable for future research.


**Reasons To Reject:**

- The paper performs OOD experiment within the US states, but more generally it would be interesting to see how well this work could transfer to other english-speaking jurisdictions.
- Line 126 states: "there has been no prior attempt to evaluate LLMs on legal IE tasks". There has been previous research work on legal IE based on LLMs, mostly on BERT-based architectures, for example: https://aclanthology.org/2023.findings-acl.187/. To the best of my knowledge there are no studies using auto-regressive LLMs such as GPT for entity extraction, there is however some work on closely related tasks such as QA or annotations, see for example: https://arxiv.org/abs/2305.04417.
- Paragraph 3.1: it would be interesting to know more about the data engineering step with OCR, since this is very common when gathering legal data and is usually a challenging task.
- Table 1 and paragraph 5: overall results are better on OOD entities and relations than there are on in domain entities and relations. The given explanation in paragraph 5 is about good transferability of the claims. This may be a partial explanation but it would be relevant to think in other directions as well: could it be that the test set is too small to accurately reflect the performance, in one direction or another? This difference in the results calls for a more precise explanation.


**Reproducibility:**

4: Could mostly reproduce the results, but there may be some variation because of sample variance or minor variations in their interpretation of the protocol or method.

**Reviewer Confidence:**

5: Positive that my evaluation is correct. I read the paper very carefully and I am very familiar with related work.

---

> ### Author Rebuttal · Authors · 2023-08-29
>
> Thank you for the review. Please find our responses to your comments and questions below.
>
> > 1. The paper performs OOD experiment within the US states, but more generally it would be interesting to see how well this work could transfer to other english-speaking jurisdictions.
>
> Thank you for the interesting suggestion. While the wills from other English-speaking jurisdictions are out of the scope of this paper, we agree that it is a great future research topic.
>
> > 2. Line 126 states: "there has been no prior attempt to evaluate LLMs on legal IE tasks". There has been previous research work on legal IE based on LLMs, mostly on BERT-based architectures, for example: https://aclanthology.org/2023.findings-acl.187/. To the best of my knowledge there are no studies using auto-regressive LLMs such as GPT for entity extraction, there is however some work on closely related tasks such as QA or annotations, see for example: https://arxiv.org/abs/2305.04417.
>
> Thank you for letting us know of the works. The first work (Barale et al., 2023) was not published yet when we submitted our paper. We will make sure to include the suggested works in our relevant works section and revise the sentence in question. Still, we’d like to emphasize that our work is novel and distinct from the suggested works. The domain and the task (Barale et al. (2023): refugee case analysis; Savelka (2023): zero-shot semantic annotation of legal texts; our work: IE from legal wills in in-context learning setting) of the works are highly distinct from each other.
>
> > 3. Paragraph 3.1: it would be interesting to know more about the data engineering step with OCR, since this is very common when gathering legal data and is usually a challenging task.
>
> We processed our collected data (in the PDF format) utilizing Python libraries called pdf2image and pytesseract. We first converted the PDF files into image files and then extracted texts using pytesseract’s “image_to_string” function. After the OCR process, the annotators manually checked the extracted texts and fixed any errors. If accepted, we will include an additional appendix with the details of this pipeline.
>
> > 4. Table 1 and paragraph 5: overall results are better on OOD entities and relations than there are on in domain entities and relations. The given explanation in paragraph 5 is about good transferability of the claims. This may be a partial explanation but it would be relevant to think in other directions as well: could it be that the test set is too small to accurately reflect the performance, in one direction or another? This difference in the results calls for a more precise explanation. How do you explain that results are significantly higher for OOD data in table 1? Same question for table 2 span difference 46% vs. 29% for OOD data: how can this difference be explained?
>
> We interpreted that the model’s good performance on OOD suggests our dataset’s high transferability. While we believe that our study has been conducted in a proper setting, we understand that there can be a concern with the relatively small sample size of test and OOD datasets. In order to mitigate this issue, we will add statistical significance analysis and confidence intervals to control for the relatively small data size when making revisions.
>
> > 5. Missing References:
> Overall no important reference is missing, but here could be a few related papers that are not cited as related work:
> Tax Law Entailment, at NLLP workshop 2020: https://arxiv.org/abs/2005.05257
> LexNLP: https://arxiv.org/abs/1806.03688
> Legal NLP review: https://aclanthology.org/2020.acl-main.466/
>
> Thank you! We will add the suggested works to the relevant works section to emphasize the large number of distinct domains in the legal NLP domain.

---

### Official Review · Reviewer_shKx · 2023-08-02

**Soundness:** 3

**Excitement:**

2: Mediocre: This paper makes marginal contributions (vs non-contemporaneous work), so I would rather not see it in the conference.

**Paper Topic And Main Contributions:**

This paper presents a new corpus for information extraction (IE) from legal wills using public domain court documents from two US states. They also evaluate the performance of LLM on this task.

**Questions For The Authors:**

Question A: Please explain the design policy of the corpus. For example, at what level of detail do you want to annotate the text?
Question B: Please explain the reason why the evaluation is performed only for four most common entity and relation types, rather than reducing the complexity of the evaluation. If there is an application that uses only these common entities and relations, it may be reasonable to have such an evaluation for the application.
Question C: Please explain the reason why you called the will from the different state is treated as out-of-domain data. I suppose the style of writing a legal will may not be that different from state to state. However, I would expect there to be a big difference for cases with different assets and different family situations. Are there significant differences in the distribution of terms or writing style in the wills from different states?

**Reasons To Accept:**

They propose a new corpus for information extraction from legal wills and confirm the possibility of building an IE system using LLM.


**Reasons To Reject:**

They do not discuss what kind of information they would like to extract from legal wills. Without such information, it is difficult to judge the value of this corpus just by the names of entities and relations. This problem is also related to the evaluation. The evaluation is done only for the four most common entities and relations. It is also difficult to interpret the evaluation results shown in the paper from the point of view of the final application process.

**Reproducibility:**

4: Could mostly reproduce the results, but there may be some variation because of sample variance or minor variations in their interpretation of the protocol or method.

**Reviewer Confidence:**

3: Pretty sure, but there's a chance I missed something. Although I have a good feel for this area in general, I did not carefully check the paper's details, e.g., the math, experimental design, or novelty.

---

> ### Author Rebuttal · Authors · 2023-08-29
>
> Thank you for the review. Please find our responses to your comments and questions below.
>
> > 1. They do not discuss what kind of information they would like to extract from legal wills. Without such information, it is difficult to judge the value of this corpus just by the names of entities and relations.
>
> Thank you for the feedback. If accepted, we will accompany the paper with an annotation manual that formalizes the taxonomy of extracted information. We will also provide concrete examples for each entity/relation category to show what information is contained in our dataset.
>
> > 2. Please explain the design policy of the corpus. For example, at what level of detail do you want to annotate the text?
>
> Our long term goal is to generate executable smart wills, and thus we annotated the information necessary for this goal as decided by the legal domain expert in our team. In order to understand, validate, and execute wills, it is crucial to extract all the key entities (e.g., testator, beneficiary, executor, witnesses, assets, just to name a few) and understand relations between entities (e.g., relations between testator and beneficiary, relations between testator and witnesses, relations between assets and beneficiary, etc.). It is also essential to extract all the key events occurring in the wills along with their arguments (e.g., in the bequest event, it is important to understand what assets (“asset” argument) are given to whom (“beneficiary” argument) in what condition (“condition” argument)). Therefore, the annotations contained in our dataset are much more detailed than those in other IE datasets constructed from other legal documents. We will accompany the paper with an annotation manual and more concrete examples if accepted, as promised earlier.
>
> > 3. Please explain the reason why the evaluation is performed only for four most common entity and relation types, rather than reducing the complexity of the evaluation. If there is an application that uses only these common entities and relations, it may be reasonable to have such an evaluation for the application.
>
> We focused on the most common entities and relations in order to mitigate the cost of generating custom prompts for each entity/relation/event type. If accepted, we will expand to include most structures that occur frequently enough in the data.
>
> > 4. Please explain the reason why you called the will from the different state is treated as out-of-domain data. I suppose the style of writing a legal will may not be that different from state to state. However, I would expect there to be a big difference for cases with different assets and different family situations. Are there significant differences in the distribution of terms or writing style in the wills from different states?
>
> There are both formal and informal differences among the wills from different states. First, obviously, wills from the different states will differ in that they must meet different legal requirements. That dictates substance, but it can also dictate style, such as the terms used for various entities, assets, and requirements. Second, practitioners and norms vary from state to state. There may be regional similarities—wills in the American South are generally more alike in style than they are when compared to ones from New Hampshire or California—but Tennessee and Idaho are not part of the same region. Plus, different case law, from different states’ courts, shapes wills as well, and states are likely to have divergence in case law.

---

### Official Review · Reviewer_Mcw9 · 2023-08-05

**Soundness:** 2

**Excitement:**

2: Mediocre: This paper makes marginal contributions (vs non-contemporaneous work), so I would rather not see it in the conference.

**Paper Topic And Main Contributions:**

The present paper proposes a dataset for information extraction on legal texts, specifically wills, from two states of the USA and some preliminary experiments using GPT-4 in-context learning for legal IE.
The authors report on the data collection and annotation process, which involved domain experts and obtained acceptable agreement scores. Further the authors evaluate GPT-4 in a in-context learning approach for identification of mentioned entities and relation extraction restricted to the four most frequent classes in the corpus for each task . The authors further evaluate the performance of the model in a out of distribution setting by processing 5 will of a different state (Idaho), while the training and test examples all consisted of will from the state of Tennessee. The authors conduct an analysis of the errors identifying that  the model overgeneralises from the prompts by identifying entities that do not belong to the desired classes - which indicates that the model does not acquire the functional aspects of n=the classes in the context of the legal texts.

**Questions For The Authors:**

- How does will processing differs essentially from legal IE in other areas, apart from specific relations, in a way that previous methods and their results could not be transferred to wills?
- Why the authors use indexes of the texts and not the tokens themselves in the entity extraction prompts? This seems to increase the difficulty for the models to learn entity descriptions.

**Reasons To Accept:**

Presents a new high quality dataset for legal IE on wills
Performs preliminary empirical studies employing LLM to legal IE in the dataset

**Reasons To Reject:**

While will processing does seem to present some interesting applications, the authors, in my opinion, fail to motivate how important the task actually is and how it differs conceptually and practically from legal IE in other areas. It is not clear to me whether the work presents a significant contribution, as the proposed resource seems very limited, with 40 wills from the state of Tennessee and five wills from the state of Idaho. I also do think that the conclusions drawn by the authors, e.g. that the model performance can be extrapolated to OOD data from the very preliminary empirical study, are unsubstantiated.

Tl;DR
The work is weakly motivated by the authors.
The empirical study is rather limited and the conclusions drawn by the authors seem unsubstantiated by their work;
The generated dataset is of very limited scope (realistically one state in the USA) and its significance is not very clear to the area

**Reproducibility:**

5: Could easily reproduce the results.

**Reviewer Confidence:**

4: Quite sure. I tried to check the important points carefully. It's unlikely, though conceivable, that I missed something that should affect my ratings.

---

> ### Author Rebuttal · Authors · 2023-08-29
>
> Thank you for the review. Please find our responses to your comments and questions below.
>
> > 1. Importance of the task (motivation for the study)
>
> Wills are important legal documents that allow people to maintain control over their assets. Unlike most legal documents, wills are commonly written by/for lay people. Given the importance and ubiquity of wills, formalizing and developing legal procedures for understanding, evaluating and executing wills is crucial. Extracting key information from legal wills is a prerequisite for understanding, evaluating and executing wills. In this vein, our task (IE from legal wills) serves as a foundation for many downstream tasks such as automatic will review and electronic will creation and execution. Also, will execution is a great use case for smart contract platforms. Unlike business contracts, court documents, including probated wills, are in the public domain in the US.
>
> > 2. Difference between IE from legal wills and from other legal documents
>
> While IE from other legal documents tend to focus on entities and simple relations, IE from legal wills need more complicated information (e.g., relations among entities, events). In order to understand, validate, and execute wills, it is crucial to understand relations between entities (e.g., relations between testator and beneficiary, relations between testator and witnesses, relations between assets and beneficiary, etc.). It is also essential to extract all the key events occurring in the wills along with their arguments (e.g., in the bequest event, it is important to understand what assets (“asset” argument) are given to whom (“beneficiary” argument) in what condition (“condition” argument).) However, existing IE datasets concentrate on extracting key entities and simple relations only. For instance, Chalkidis et al. (2017) extracts various contract elements such as contract title, dates, relevant law articles, but it doesn’t extract any relations among such elements nor extract events from the document. Chen et al. (2020) does extract four relations relevant to key criminal behaviors, but the level of sophistication is different from our work. Due to this difference, it is difficult to transfer IE framework from other legal domains to handle IE from legal wills.
>
> > 3. Limited scope of the dataset
>
> It is correct that the scope of our dataset is somewhat narrow. However, we’d like to point out that our work is a continued effort and thus the scope of our dataset will continue to expand. Also, our experiment with LLM suggests that the knowledge from our dataset can be transferred to handle OOD data. This shows that our dataset can be applied to broader domains than its direct scope.
>
> > 4. The conclusions drawn by the authors, e.g. that the model performance can be extrapolated to OOD data from the very preliminary empirical study, are unsubstantiated.
>
> The conclusion that the model’s performance can be extrapolated to OOD data is based upon our findings from the experiment. The decent F1 scores from OOD data (79% for entities and and 83% for relations) suggests that the knowledge from our dataset can be successfully transferred to other domains (e.g., wills from other US States). While we believe that our study has been conducted in a proper setting, we understand that there can be a concern with the relatively small sample size of test and OOD datasets. In order to mitigate this issue, we will add statistical significance analysis and confidence intervals to control for the relatively small data size when making revisions.
>
> > 5. Why do the authors use indexes of the texts and not the tokens themselves in the entity extraction prompts?
>
> We’d like to clarify that we used both the actual tokens and the indices in the entity extraction prompts. For example, when extracting the text “I” as an entity, we provided both the text “I” and start and end indices (69 and 70, respectively) to the model. The reason we utilized both the texts and their indices in the prompt is to distinguish between the tokens that have the same texts (e.g., The reference of pronouns such as “his”, “her”, or “their” can be confusing if there are many candidates in the text).

---

### Meta-Review · Area_Chair_7t5M · 2023-09-17

**Recommendation:** 3

**Metareview:**

The paper contributes a new high-quality dataset for legal Information Extraction on legal wills derived from public domain court documents from two US states. It discusses preliminary empirical studies to investigate the capability of ChatGPT4 on the task.  Reviewers and authors exchanges during discussion helped clarify some of the critical issues highlighted in the reviews. Overall, the paper was found to be interesting and innovative. However, some revisions are needed to make it more convincing. For instance, discussing the differences between the wills from the two states, as mentioned in response to the reviewers, would yield valuable insights for other researchers.  Given that this is a short paper, it needs at least to better explain the importance and novelty of IE on legal wills and to better justify the design choices by providing additional details, as indicated by reviewers.

Here below is a summary of the main strengths and weaknesses identified:

**Pros**

- a new corpus for information extraction from legal wills;

- demonstration  of the adequacy of the Gpt4 model for IE tasks;

- focuson information extraction and linking, of special interest in legal profession;

- comparison with other domains, interesting for highlighting shared challenges;

- portability to other US states legal wills;

- presence of error analysis.

**Cons**

- the dataset is limited, as it covers few states and few cases. Increasing its coverage will certainly provide stronger support to the claims;

- the importance and novelty, or difference from similar tasks in legal IE, is not evident. The paper should explain it more clearly;

- details on the extracted and evaluated information categories seems to be missing, which leaves some of the choices unexplained.

---

### Decision · Program_Chairs · 2023-10-07

**Decision:**

Accept-Findings

**Comment:**

The paper contributes a new high-quality dataset for legal Information Extraction on legal wills derived from public domain court documents from two US states. It discusses preliminary empirical studies to investigate the capability of ChatGPT4 on the task.  Reviewers and authors exchanges during discussion helped clarify some of the critical issues highlighted in the reviews. Overall, the paper was found to be interesting and innovative. However, some revisions are needed to make it more convincing. For instance, discussing the differences between the wills from the two states, as mentioned in response to the reviewers, would yield valuable insights for other researchers.  Given that this is a short paper, it needs at least to better explain the importance and novelty of IE on legal wills and to better justify the design choices by providing additional details, as indicated by reviewers.

Here below is a summary of the main strengths and weaknesses identified:

**Pros**

- a new corpus for information extraction from legal wills;

- demonstration  of the adequacy of the Gpt4 model for IE tasks;

- focuson information extraction and linking, of special interest in legal profession;

- comparison with other domains, interesting for highlighting shared challenges;

- portability to other US states legal wills;

- presence of error analysis.

**Cons**

- the dataset is limited, as it covers few states and few cases. Increasing its coverage will certainly provide stronger support to the claims;

- the importance and novelty, or difference from similar tasks in legal IE, is not evident. The paper should explain it more clearly;

- details on the extracted and evaluated information categories seems to be missing, which leaves some of the choices unexplained.